# Smooth Muscle Cells from a Rat Model of Obesity and Hyperleptinemia Are Partially Resistant to Leptin-Induced Reactive Oxygen Species Generation

**DOI:** 10.3390/antiox12030728

**Published:** 2023-03-16

**Authors:** Ocarol López-Acosta, Magdalena Cristóbal-García, Guillermo Cardoso-Saldaña, Karla Carvajal-Aguilera, Mohammed El-Hafidi

**Affiliations:** 1Depto. de Biomedicina Cardiovascular, Instituto Nacional de Cardiología Ignacio Chávez, Juan Badiano No 1, Colonia Sección XVI, Tlalpan 14080, Mexico; ola_runi@hotmail.com (O.L.-A.); magdalena.cristobal@cardiologia.org.mx (M.C.-G.); 2Depto. de Endocrinología, Instituto Nacional de Cardiología Ignacio Chávez, Juan Badiano No 1, Colonia Sección XVI, Tlalpan 14080, Mexico; cargui@cardiologia.org.mx; 3Laboratorio de Nutrición Experimental, Instituto Nacional de Pediatría, Insurgentes Sur 3700, Col. Insurgentes Cuicuilco, Coyacan 4570, Mexico; kcarvajala@pediatria.gob.mx

**Keywords:** leptin, oxidative stress, smooth muscle cell, metabolic syndrome and antioxidant defense

## Abstract

The aim of this study was to evaluate the effect of leptin on reactive oxygen species’ (ROS) generation of smooth muscle cells (SMCs) from a rat model of obesity and hyperleptinemia. Obesity and hyperleptinemia were induced in rats by a sucrose-based diet for 24 weeks. ROS generation was detected by using dichloro-dihydrofluorescein (DCF), a fluorescent ROS probe in primary SMCs culture. An increase in plasma leptin and oxidative stress markers was observed in sucrose-fed (SF) rats. At baseline SMCs from SF rats showed a more than twofold increase in fluorescence intensity (FI) compared to that obtained in control (C) cells. When the C cells were treated with 20 ng leptin, the FI increased by about 250%, whereas the leptin-induced FI in the SF cells increased only by 28%. In addition, sucrose feeding increased the levels of p22phox and gp91phox, subunits of Nox as an O_2_^•−^ source in SMCs. Treatment of cells with leptin significantly increased p22phox and gp91phox levels in C cells and did not affect SF cells. Regarding STAT3 phosphorylation and the content of PTP1B and SOCS3 as protein markers of leptin resistance, they were found to be significantly increased in SF cells. These results suggest that SF aortic SMCs are partially resistant to leptin-induced ROS generation.

## 1. Introduction

Metabolic syndrome (MetS) and obesity are risk factors for the development of cardiovascular diseases (CVD) such as hypertension, atherosclerosis and heart failure, which are considered leading causes of death worldwide [1]. In obesity associated with hyperleptinemia, leptin-induced reactive oxygen species (ROS) generation in vascular tissues is suggested to be a mechanism involved in the physiopathological processes of hypertension and atherosclerosis [2,3,4,5]. Leptin, a 16-kDa polypeptide, was primarily reported to be only released from adipose tissue to regulate food intake energy expenditure in the brain [6,7]. However, cardiomyocytes and vascular smooth muscle cells (SMCs), were recently described as both producers and targets of leptin [8,9,10]. Leptin exerts its biological action by binding to its receptor, which is expressed in several peripheral tissues [11] but its physiological function remains to be established. Various peripheral tissues, including liver and lung, express the short-form leptin receptor (Lep-Ra) which is only involved in leptin transport [12]. In the vasculature, the long-form leptin receptor (Lep-Rb), with a molecular size of 170 kDa, is expressed in endothelial cells [13]. In vascular smooth muscle cells (SMC), the 130 kDa short form of the leptin receptor was reported to mediate the proliferative phenotype [14]. The effect of leptin on artery relaxation is controversial. In some reports, leptin was found to impair vascular relaxation via its receptor and induce superoxide anion generation which reduces nitric oxide availability [15,16]. However, other reports have shown that leptin protects against aorta contractile response to angiotensin II, either by inhibiting the increase in cytosolic Ca^2+^ or by a mechanism involving nitric oxide generation [17,18]. In endothelial cells and cardiomyocytes, leptin increases the level of oxidative stress markers [19], participates in vascular remodeling, since it induces rat aortic SMC proliferation [14], and promotes SMC neo-intimal growth after vascular injury in mice [20]. Leptin-induced SMC proliferation is mediated by ROS generation via NADPH oxidase (Nox) assembly and activation [21]. Nox activity is considered the most important source of superoxide anion (O_2_^•^^−^) in vascular tissues and plays an important role in vascular remodeling and dysfunction during the development of CVD [22,23].

In pathological conditions such as obesity and MetS, several circulating factors such as leptin can directly interact with vascular cells to induce a change in cell phenotype and damage by a mechanism that involves ROS generation. Therefore, most studies on leptin effect have been performed on normal cells or cells exposed to angiotensin to mimic impaired relaxation or cell proliferation [24]. Studies using cells from obese or hyperleptinimic models to understand the pathophysiological role of hyperleptinemia on ROS generation are rare. Our group experimented on a model of intra-abdominal fat accumulation induced by a high sucrose diet where aorta ring vasoconstriction was found to be enhanced in response to norepinephrine, and vasorelaxation reduced in response to acetyl-choline [25]. These responses are associated with increased ROS generation through Nox activity in the vascular tissue [25]. Recently, a significant structural change of the middle medium layer of the thoracic aorta of SF animals compared to C rats was reported [26]. A sucrose diet induces a change in vascular tissue thickness that may be associated with a change in the contractile to synthetic (proliferative) smooth muscle cell phenotype that might be associated with alterations in aortic ring vascular reactivity and hypertension. In addition, the model of sucrose-diet induced obesity is characterized by increased plasma free fatty acid (FFA) levels, hypertriglyceridemia (TG) and hyperleptinemia, and is expected to develop leptin resistance [27]. Indeed, lowering plasma leptin levels, either by inhibiting adipocyte leptin production or by increasing kidney leptin clearance, has been described to restore and improve leptin sensitivity in mice [21,28].

The interaction of leptin with its Lep-R involves the participation of protein tyrosine phosphatase-1B (PTP1B), the suppressor of cytokine signaling 3 (SOCS3), and of the signal transducer and activator of transcription 3 (STAT3), well-known molecules that attenuate leptin signaling [29]. For example, SOCS3 is a leptin-inducible inhibitor of leptin signaling. Therefore, the use of SMCs extracted from an obese model with hyperleptinemia, may be a relevant approach for studying the leptin action and the mechanism by which leptin contributes to ROS generation. Hence, the objective of this study was to evaluate the effect of leptin on ROS generation and on the antioxidant system, in a primary culture of SMC derived from the aorta of a model of sucrose-diet induced obesity.

## 2. Experimental Design

*Animals.* The experiments were conducted in compliance with the Mexican Federal Regulation for Animal Experimentation and Care (NOM-062-ZOO-2001). Weanling male Wistar rats aged 4 weeks and weighing approximately 65 ± 5 g were obtained from the animal facility of the National Institute of Cardiology Ignacio Chávez. The animals were divided into two groups of 12 rats each: the control group (C) received a solid food ad libitum (Lab diet formula 5001, Ralston Purina Corp., St Louis, MO, USA) and water; the sucrose-fed group (SF) received a 30% sucrose solution in drinking water and the same solid food ad libitum as the C group. At the end of the treatment period, blood pressure measurements of the rats were performed by the tail-cuff method: the cuff was connected to a pneumatic pulse transducer (Narco Bio-Systems, a Healthdyne Co., Houston, TX, USA) and a programmed electro-sphygmomanometer from the same company. Recordings were obtained in duplicate by means of a Narco Bio-Systems polygraph [30]. After 24 weeks of treatment, the rats were fasted overnight and sacrificed the next day.

*Plasma leptin, insulin, FFA, TG, and glucose analysis.* Blood was collected from the abdominal aorta into tubes containing an anticoagulant (0.1% EDTA) and immediately centrifuged at 600× *g* for 20 min at 4 °C. To the plasma thus obtained, 0.005% of butylated hydroxy toluene (BHT) was added as an antioxidant and the mixture was stored at −70 °C until FFA analysis, which was performed by gas chromatography as described previously [31] while lipids were extracted according to the method of Folch et al. [32]. Plasma glucose and TG concentrations were measured according to the method described by Nagele et al. [33]. Plasma insulin and leptin levels were measured using an insulin and a leptin kit (Linco Research, St. Charles, MO), respectively. The HOMA-IR was calculated from the insulin and glucose values using the following formula: {[insulin] (in mU/l) × [glucose] (in mmol/l))/22.5}. Intra-abdominal fat was dissected off retroperitoneal cavity and around both kidneys, and immediately weighed. Visceral and duodenal fat was not included in this procedure.

*Oxidative stress markers*. Plasma carbonyl proteins were quantified in plasma prepared as described above using a modified method as reported previously [34,35]. Thiobarbituric acid reactive substance (TBARS) was determined in the plasma as described previously [30].

*Smooth muscle cell primary culture.* The SMCs were extracted from the aorta in sterile conditions as described previously [36]. The tissue was immediately placed in a buffer containing 140 mM NaCl, 4.7 mM KCl, 1.2 mM Na_2_HPO_4_, 2.4 mM MgSO_4_, 2 mM CaCl_2_, 5.6 mM glucose, 0.02 mM EDTA, 25 mM HEPES, pH 7.4.

In a first step, the fat was removed and the aorta was incubated with 1 mg/mL type II collagenase (Gibco) for 20 min to discard the adventitial tissue from smooth muscle tissue along the aortic wall. In a second step, aorta without adventitia was incubated in the medium containing 1 mg/mL papain (Roche) for 40 min to disperse the smooth muscle cells. Subsequently, the SMCs were filtered through a 230 μm pore sieve (Tissue Grinder Kit, Sigma) and seeded in a 25 cm^2^ culture flask, in DMEM/F-12 as the culture medium (Gibco), supplemented with 10% fetal bovine serum (FBS, Gibco) which was inactivated at 56 °C for 30 min. The SMCs were incubated at 37 °C, in an atmosphere of 5% CO_2_ and 90% humidity. When the cells reached confluence, they were dispersed with 0.25% trypsin-EDTA (Gibco) and re-seeded in a 75 cm^2^ tissue flask in the same culture medium as described above (first passage).

*Smooth muscle cell ROS generation.* In the second passage, cells were seeded at 10,000 cells per cm^2^ for rapid growth in a 6-well cell culture plate (Corning Incorporated) containing a DMEM/F12 medium supplemented with 10% FBS. After 48 h of culture, cells were incubated in the same medium without FBS but in the presence of 10 μM of 2′,7′ dichlorodihydrofluorescein diacetate (DCF-DA) for 15 min. After washing the cells from excess DCF, they were stimulated for a further 16 min with leptin at different concentrations (20, 40 and 80 ng/mL culture mediums) to induce ROS generation. At the end of the experiment, the cells were washed and fixed with 1% paraformaldehyde for 30 min at 4 °C. To visualize the nucleus, the fixed cells were incubated with 5 μM DAPI (4′,6-diamidino-2-phenylindole).

The effect of leptin on ROS generation was also analyzed in the presence of apocynin and diphenylene-indol (DPI) as Nox inhibitors, or in the presence of N-acetyl-cysteine (NAC) as a reducing agent. The assay was performed on adhered cells grown on a 6-well culture flask that was pre-incubated for 15 min with apocynin, DPI or NAC, and then incubated with 10 μM DCF-DA for 15 min before the cells were stimulated with leptin at 40 ng for 15 more minutes to stimulate ROS generation. The cells were then washed with PBS to remove excess DCF-DA and were fixed with 1% paraformaldehyde for 30 min at 4 °C. The cells were then washed again three times with PBS. The fluorescence of oxidized DCF and DAPI was detected by fluorescence microscopy.

*Image analysis.* Images were obtained by fluorescence microscopy using LSM-700 Zeiss equipment (Baden-Württemberg) equipped with a 20X objective. The fluorescence of oxidized DCF and DAPI were detected at Excitation326nm/Emission432nm and Ext358/Em461, respectively. The fluorescence image was analyzed using SEISS ZEN microscope software. The fluorescence intensity (FI) of DCF in a given area was divided by the FI of DAPI as a reference of the number of nuclei. The ratio of the FI (DCF)/FI (DAPI) value was reported in the results.

*Effect of leptin on protein content.* As described above, in the second passage, SMCs were seeded at 10,000 cells per cm^2^ for rapid growth in a DMEM/F12 medium, supplemented with 10% FBS in Corning (100 × 20 mm) style dishes. When the cells reached confluence, they were stimulated with leptin for 24 h. At the end of the experiment, cells were harvested and lysed in a buffer containing 100 mM Tris-HCl, 5 mM sodium pyrophosphate, 10 mM EDTA (pH 7.2), 50 mM NaF and 1% Triton X100, and supplemented with 1 mM sodium orthovanadate, 1 mM phenylmethylsulfonyl-fluoride (PMSF), 2 μg/mL aprotinin, 2 μg/mL pepstatin and 2 μg/mL leupeptin as anti-proteases. The sample was then centrifuged at 8000× *g* for 10 min at 4 °C to remove cell debris. Protein levels were quantified in the homogenate using the Bradford method [37]. Two hundred μg protein of each sample was collected, suspended in 25 μL of buffer load containing 125 mM Tris-HCl (pH6.8), 20% glycerol, 4% SDS, 10% 2-mercaptoethanol and 0.004% bromophenol blue, completed to 50 μL with Laemmli’s solution (40 mM Tris, 1% SDS and 1% β-mercaptoethanol).

Fifty to 80 μg protein of sample was loaded into an SDS-PAGE gel with the acrylamide percentage depending on the protein to be analyzed as indicated in the figure legend. The electrophoresis was run for 3 h at 120 V. The protein transfer was performed onto a PVDF (polyvinylidene fluoride) membrane with a pore size of 0.22 μm (Immobilon Millipore) at 350 mA for 60 min in a semi-dry transfer chamber (Bio-Rad, Trans Blot SD). The non-specific protein detection was reduced by blocking membranes in a TBS (25 mM Tris, 150 mM NaCl) solution containing 5% skim milk and 0.1% Tween 20. Then, the membranes were incubated with polyclonal antibodies against (Cu/Zn)SOD, (Mn)SOD, catalase, p22phox, gp91pox, Nox4, leptin receptor (Lep-R) and GAPDH as a control load from Santa Cruz (Biotechnology Corporation, Santa Cruz, CA, USA) except for the anti-catalase antibody, anti-SOCS3, anti-PTP1B, anti-STAT3 and phosphorylated anti-STAT3, which were purchased from Abcam. The secondary antibodies used were peroxidase conjugated. Proteins were revealed by chemiluminescent reagent clarity (Bio-Rad) and the membranes were exposed to image plates (BioMax, Kodak) for 5 min. The image plate was captured with an imaging system GelDoc-It (UVP Inc., Upland, CA, USA). Bands were analyzed by a UVP image analyzer and optical density (OD) was evaluated with VisionWorks LS software (UVP Inc., Upland, CA, USA).

*The SOD and CAT activities.* The activities of SOD and CAT in vascular SMC homogenate were assessed using the technique of native polyacrylamide gel staining, which allows the assay of the activity of CAT or single SOD isoform and excludes interference from non-CAT or non-SOD molecules in the crude tissue extract, which is not possible to avoid when using the spectrophotometric method [38]. Briefly, 10 μL of each sample containing 2 mg/mL total protein was loaded onto 5% staking and 8% native polyacrylamide gel (without SDS) and the proteins were separated at constant current (120 V) at 4 °C for 3 h. After electrophoresis, the gels were washed with a 50 mM phosphate buffer (pH: 7.8) for 10 min and then incubated in a solution containing 50 mM potassium phosphate (pH: 7.8), 275 μg/mL nitroblue tetrazolium (NBT), 65 μg/mL riboflavin and 0.25% tetramethylenediamine (TEMED). After 15 min incubation in the dark, the blue NBT stained gel for O_2_^•^^−^ was rinsed in phosphate buffer and illuminated for 15 min with a UV light source. The SOD activity appeared as clear bands on a purple background.

For CAT activity, the washed gel was incubated in a buffer containing 5 mM H_2_O_2_ for 10 min and excess H_2_O_2_ was eliminated by washing the gel three times with double distilled water. Then the gel was stained with a mixture of FeCl_3_/K_3_Fe(CN)_6_ (potassium ferricyanide) at 1% each. The CAT activity appears as clear bands on a blue-green background. Then, the gels were immediately captured with a GelDoc-It imaging system and analyzed by a UVP image analyzer as described above. The results were reported as pixels.

## 3. Statistical Analysis

All statistical analyses were performed with Sigma plot version 11 (Systat Software Inc., San Jose, CA, USA). All values are expressed as means ± SD. Differences between groups were analyzed by one-way ANOVA for selected variables, followed by a Tukey ad hoc test. The number of animals used for each analysis is indicated in the figure and table legends. Statistical differences were considered significant when *p* < 0.05.

## 4. Results

### 4.1. General Characteristics of Animals

The treatment of rats with 30% sucrose in drinking water for 24 weeks induced a statistically significant increase (*p* < 0.05 to *p* < 0.01) in systolic blood pressure, plasma TG, FFA, insulin and leptin, and a greater accumulation of intra-abdominal adipose tissue (Table 1). No significant changes in body weight, glucose and total cholesterol were observed between the groups (Table 1). However, the significant difference in HOMA-IR between the control and SF rats indicates insulin resistance in the model, as previously demonstrated by the hyperinsulinemic/euglycemic clamp experiment [35]. Regarding cholesterol associated with HDL, a significant decrease was observed in the SF animals (*p* < 0.05).

### 4.2. Oxidative Stress Markers

Protein carbonyls, markers of oxidative stress in vivo, were significantly increased by approximately 50% (*p* < 0.05) in the plasma of the SF animals (Table 1). In regard to TBARS, a marker of lipid peroxidation, it was found significantly enhanced in the plasma of SF rats by approximately 50% (*p* < 0.05).

### 4.3. Smooth Muscle Cell ROS Generation

To evaluate ROS generation by SMCs, we used DCF-DA, which is permeable to the plasma membrane and, once in the cell, is hydrolyzed to free DCF through the action of a non-specific esterase and is oxidized by ROS. Microscopic images (Figure 1) show that in DCF-DA loaded SMCs, the fluorescence intensity at baseline (without leptin stimulation) was three and a half times more intense in SF cells compared to cells from control animals. Figure 1 shows the result obtained from the ratio of DCF FI to DAPI FI as a reference to cell number. The bottom panel also shows the effect of leptin on the ratio in a dose-dependent manner in control cells with an increase of approximately 250%, 425% and 700% at 20, 40 and 80 ng of leptin, respectively. Whereas leptin enhanced the ratio in SF cells only by 28%, 44% and 52% at 20, 40 and 80 ng, respectively (Figure 1).

The increase in FI induced by 20 ng leptin was modulated in cells from control and SF rat aortas by apocynin and DPI, Nox inhibitor, and by NAC as a reducing agent (Figure 2). The increased fluorescence of DCF induced by leptin was more sensitive to apocynin and DPI than to NAC (Figure 2).

Both apocynin and DPI inhibit the FI of oxidized DCF induced by leptin in a dose-response manner in C and SF cells. The ratio of FI (DCF)/FI (DAPI) was decreased by 60% in both SF and control cells (Figure 2) by DPI at 10 μM.

The presence of apocynin, a more selective inhibitor of Nox, when pre-incubated with SMC, also inhibited the increase in the FI ratio in a dose-response manner (Figure 2) and decreased ROS generation in SF cells but with less efficacy than did DPI. In regard to NAC, a reduction of the ratio was observed but without reaching a statistically significant difference.

### 4.4. Western Blot of Nox and Their Subunit Proteins

The Western blot analysis showed a significant increase in the level of p22phox, by approximately 170% in SMCs from SF aortas as compared with C (Figure 3a). When leptin was incubated with the cells for 24 h, the p22phox content was found to be enhanced in the control cells by approximately 64% and was not significantly affected in SF cells. Concerning the gp91phox content (Figure 4b), a significant increase in the protein level was observed in SMCs from the SF aortas in comparison with the control SMCs.

When cells were incubated with leptin, a slight but a significant (*p* < 0.05) increase in gp91phox was observed in control cells. In SF cells however, leptin did not significantly affect the gp91phox level (Figure 3b). In regard to Nox4 content, no significant difference in the protein level was observed between SF and control cells (Figure 3c) and no significant change was observed under the effect of leptin in SF and control SMCs (Figure 3c).

### 4.5. Leptin Receptor (Lep-R)

The Western blot analysis demonstrated the presence of a band that matched the predicted molecular weight of the Lep-R form expressed in the brain (see supplemented data). Sucrose feeding induced a significant decrease in receptor levels compared with the control cells. When cells were incubated with 20 ng/mL leptin for 24 h in a primary culture, the band was significantly decreased in the control cells and slightly enhanced in SMCs from SF rats (Figure 4a).

### 4.6. Effect of Leptin on STAT3, PTP1B and SOCS3

The Western blot analysis showed a significant increase in the level of PTP1B by approximately 20% in SF smooth muscle cells as compared with the control cells (Figure 4c). In regard to leptin treatment (20 ng), this did not affect the content of PTP1B in either the control cells or the SF cells. Concerning SOCS3 levels (Figure 4b), a significant difference between C and SF smooth muscle cells was observed. When cells were incubated with leptin, a moderate increase of SOCS3 was observed in both C cells but no effect was observed in the SF cells.

The Western blot analysis of STAT3 phosphorylation shows an increase in the content of phosphorylated STAT3 in cells from SF animals and did not depend on leptin treatment. With regard to total STAT3 content, no significant difference was observed between C and SF cells, nor with the leptin-treated cells (Figure 5).

### 4.7. Effect of Leptin on Catalase and SOD Contents and Activities

The effect of sucrose feeding and leptin treatment of SMCs on CAT and SOD content and activities involved in H_2_O_2_ degradation and O_2_^•−^ to H_2_O_2_ dismutation, respectively, were also assessed by native polyacrylamide gel staining. Figure 6a and the corresponding native gel show a significant increase in CAT activity in SF cells compared with the control cells, and leptin did not significantly affect CAT activity in either the SF cells or control cells. Regarding CAT content, Western blot showed no difference between control and SF aortic SMCs or after leptin treatment (Figure 6b). Similarly, Cu/Zn-SOD activity was significantly increased by 50% in SF smooth muscle cells compared with C cells and leptin treatment did not affect this activity (Figure 6c).

The Western blot analysis of Cu/Zn-SOD showed no difference in the protein content of this enzyme between the SF and control cells (Figure 6d), and leptin treatment did not affect its expression in the SMCs of both groups of animals. In the presence of leptin, the protein content showed no change in the cells compared to untreated cells (Figure 6d). For Mn-SOD, which is specific to mitochondria, a native polyacrylamide gel staining analysis showed no difference between the C and SF cells and leptin treatment decreased the activity of the enzyme in both C and SF cells (Figure 6e), whereas the amount of protein did not change either between C and SF cells or under leptin treatment (Figure 6f).

## 5. Discussion

This work was undertaken to investigate the ability of leptin to induce in vitro ROS generation in SMCs obtained from the aortic tissue of SF and C rats, and also to show that in vitro research using a primary culture of SMCs, may closely mimic the in vivo response of SMCs to hyperleptinemia-induced ROS generation. Sucrose feeding induces hyperleptinemia, oxidative stress markers and changes in several clinical parameters such as blood pressure, intra-abdominal obesity and hypertriglyceridemia which reflect several characteristics of the clinical diagnosis observed in patients with obesity and metabolic syndrome. Therefore, these findings make the animal model of sucrose feeding useful to investigate the smooth muscle cell ROS generation and oxidative stress as a mechanistic link between obesity and the development of cardiovascular diseases, even though there was no weight gain in the sucrose feeding model. In a previous work, we described how the lack of weight difference between the control and SF groups is attributable to the lack of difference in their energy intake. Indeed, SF rats ingested less solid food than the control animals [39]. Body composition was not examined in detail in this study, and we do not know whether sucrose administration caused an increase in adipose tissue content at the expense of other tissues. The SF rats consumed approximately half the amount of food consumed by animals that were not given sucrose; consequently, the availability of nutrients from the solid food was lower as described previously [39]. Thus, the lower energy intake of the SF rats was compensated by the additional calories from the sucrose solution.

The in vitro increase in ROS generation in a primary culture of SF rats’ SMCs can be attributed to enhanced Nox expression or activity induced by a sucrose-rich diet during the animal treatment period. Nox is a multiprotein complex composed of different subunits. Among them, the p22phox subunit, one of the most important subunits because of its catalytic site, generates O_2_^•−^ and gp91phox, which are required for Nox complex assembly and activity in cell membrane: hence an increase of the subunit levels in SF smooth muscle cells is related to the increased O_2_^•−^ generation. In the SF rat model, enhanced SMC ROS generation and oxidative stress markers may be considered as a consequence of obesity, and metabolic alterations induced Nox activity and expression as described previously [25]. However, it was reported that the overexpression of p22phox, a subunit of Nox, in vascular SMCs in mice, increased vascular ROS production, caused obesity and increased fat mass [40]. This observation allows us to speculate that increased ROS generation in SMCs may contribute to the development of obesity with hyperleptinemia and a possible leptin resistance in our SF experimental model. Indeed, chronic exposure of vascular tissue to high levels of circulating leptin can induce leptin resistance in SF rats, as described elsewhere [41,42]. In SMCs from SF animals, the low response to leptin-induced ROS generation of the Nox subunits p22phox and gp91phox levels as compared with control SMCs suggests leptin resistance in the cell. When 20 ng/mL leptin was added to control cells, FI increased by 225% and only by 28% in SF aorta cells. In addition, leptin-induced ROS generation through Nox activity was evidenced by treating SF and control SMCs with both DPI and apocynin, widely used in the literature to elucidate the involvement of Nox in different biological systems. The involvement of Nox in ROS generation in SMCs can be attributed to the interaction with leptin and its receptor Lep-R as described in endothelial cells 42 [40]. In phagocytic cells, Lep-R was described to be coupled to Nox activity involved in ROS generation [43]. Evidence for the direct involvement of the leptin receptor in ROS generation will be further investigated in our laboratory. However, the result of the Western blot analysis shows that the level of Lep-R in SF does not differ from that in the control SMCs, suggesting an equal involvement of the leptin receptor in both cell types. Lep-R signaling initiated by the interaction of leptin with its receptor, leads to phosphorylation of the tyrosine residue of the receptor by activation of Jak 2. This mediates different signals, such as phosphorylation of STAT3, which activates transcription of the SOCS3 gene. After long-term stimulation, the SOCS3-translated protein binds to the phosphorylated Tyr of Lep-R and thus inhibits Lep-R-mediated signals [44].

In this study, the increased SOCS3 and PTP1B level in SMCs from SF is related to an increase of STAT3 phosphorylation, suggesting an alteration in leptin signaling in SF smooth muscle cells in response to leptin-induced ROS generation. Indeed, the over expression of the constitutively active form of STAT3 (phospho-STAT3) translocates to the nucleus and activates the SOCS3 which results in leptin resistance and fat accumulation in mice [45]. Our findings suggest that sucrose feeding induces phosphorylated STAT3 and SOCS3 as a leptin-inducible inhibitor of leptin signaling and blocks leptin-induced signal transduction in SMCs.

Hence, hyperleptinemia and high FFA from increasing the STAT3 phosphorylation in SMCs impairs the leptin signaling pathways during the development of diet-induced obesity, which is associated with disorders of energy homeostasis due to diet-induced obesity [46,47]. Therefore, excessive SOCS3 activity is considered as a potential mechanism for the leptin resistance that characterizes human obesity. Other intracellular proteins such as PTP1B provide a negative feedback regulatory mechanism to prevent over-activation of Lep-R pathways. In db/db mice, a model of diabetes, the deletion of PTP1B was found to improve leptin resistance and reduce superoxide generation [48]. In addition, leptin receptor-deficient ob/ob and db/db mice develop cardiovascular and vascular dysfunction [49]. These leptin and leptin receptor-deficient rodent models have provided many useful insights into the underlying molecular and pathophysiological mechanisms of metabolic and cardiovascular diseases associated with obesity and type 2 diabetes. For example, it has been described that animals deficient in leptin receptors show elevated triglyceride levels and lipid accumulation in the myocardium, which may promote lipotoxicity and directly impacts cardiac contractility [50].

The balance between ROS generation and the antioxidant system to maintain ROS at physiological levels contributes to the normal endothelial function and smooth muscle cell contraction in the vascular system. A loss of this balance results in the uncontrolled production of ROS leading to the development of cardiovascular diseases [51]. Under conditions of oxidative stress induced by high ROS generation, the increased expression of antioxidant enzymes has been described in skeletal muscle [52]. The increased ROS generation in SF smooth muscle cells can also be due to the increased dismutation of superoxide anion to H_2_O_2_ by increased activities of Cu/Zn-SOD and Mn-SOD in both cytosol and mitochondria, respectively. This increase in the activities of Cu/Zn-SOD, Mn-SOD and catalase in SMCs from SF rats, can also be considered as a protection against excessive superoxide anion and H_2_O_2_ generation. However, the endogenous oxidant H_2_O_2_, depending on the concentration, can be considered of central importance in redox signaling to modulate cellular redox status including enzyme activities. At elevated concentrations, ROS can directly modulate the activity of SOD by a reaction with the catalytic site and some amino acid residues, inducing changes in protein conformation and antioxidant activity. In the control aortic SMCs, the leptin-reduced effect of Mn-SOD activity can be indirectly related to its ROS-inducing effect via activation of NADPH oxidase and mitochondria [53,54]. To our current knowledge, there is no data about the direct effect of leptin on the activity of antioxidant enzymes (catalase, Cu/Zn SOD and Mn-SOD). Therefore, further studies are needed to elucidate the mechanism by which leptin modulates the antioxidants’ enzyme activity in SMCs from both the control and SF animals.

What is known to date is that vascular hypertrophy was blunted in SOD1^−/−^ mice compared to WT mice [55] and that Cu/Zn-SOD contributes to visceral fat accumulation by causing insulin secretion and insulin resistance by a mechanism that involves ATP production in mitochondria [56]. In regard to catalase, its overexpression was also described to inhibit SMC proliferation by reducing the content of endogenous and exogenous H_2_O_2_ [57]. Hydrogen peroxide has been described as acting as a second messenger by modulating the activity of kinases and phosphatases in several signaling pathways to induce cell proliferation, differentiation and migration involved in the progress of vascular disease [58,59]. In addition, apocynin reduces superoxide anion generation via Nox inhibition as described previously [36].

In summary, our results indicate that SF rats with hyperleptinemia and high fat accumulation, the coexistence of which is a marker of leptin resistance, and increased leptin protein signaling may partly explain the differential effect of leptin-induced ROS generation through Nox activities in SF and C smooth muscle cells.

Taken together, these models have been widely used for various purposes, and future research will address in detail the molecular basis and the pathophysiological manifestations of these models as well as how they compare with human T2DM disease states.

## Figures and Tables

**Figure 1 antioxidants-12-00728-f001:**
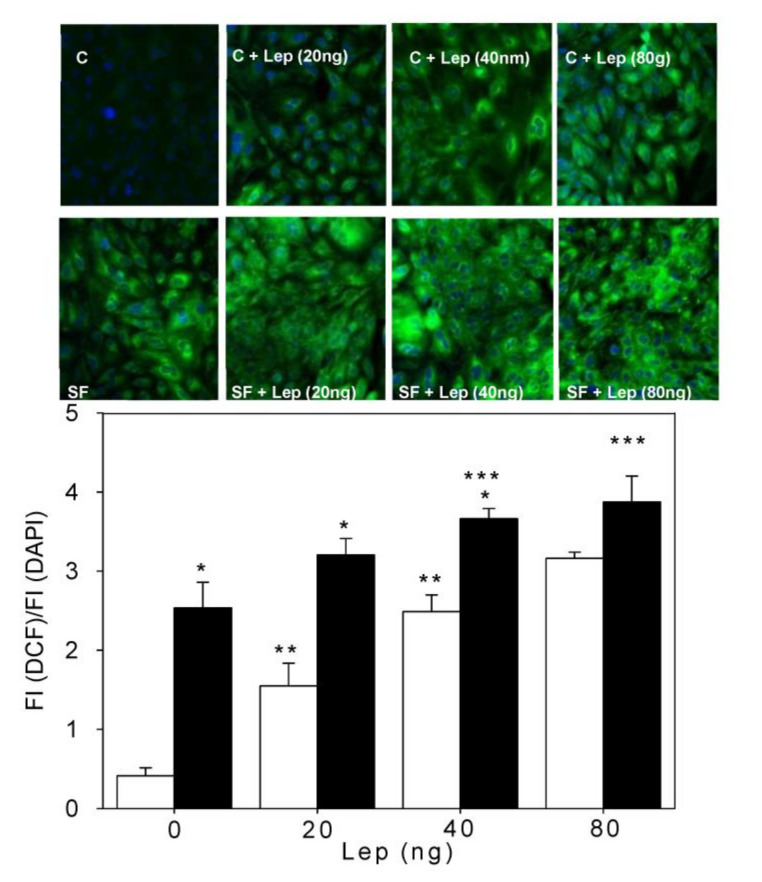
Leptin and ROS generation in SMCs. Aortic SMCs from both the control (C) and sucrose-fed (SF) rats were loaded with 10 μM 2,7-dichlorodihydrofluorescein diacetate (DCF-DA) and treated with leptin at different concentrations (from a 20 to 80 ng/mL culture medium) followed by 5 μM DAPI (Blue) to mark the nucleus, as described in the Method section. The FI of oxidized DCF (green) was detected using fluorescence microscopy equipped with a 20X objective (see images). All images were processed in the same conditions and using the same microscope parameter to determine fluorescence intensity. The graph presents the ratio of DCF FI to DAPI FI +/− SD (n = 6 different experimental animals). The white bars correspond to the control SMCs and the black bars correspond to SF SMCs. * *p* < 0.01 corresponds to C vs. SF; ** *p* < 0.01corresponds to C vs. C + Lep and *** *p* < 0.05 corresponds to SF vs. SF + Lep.

**Figure 2 antioxidants-12-00728-f002:**
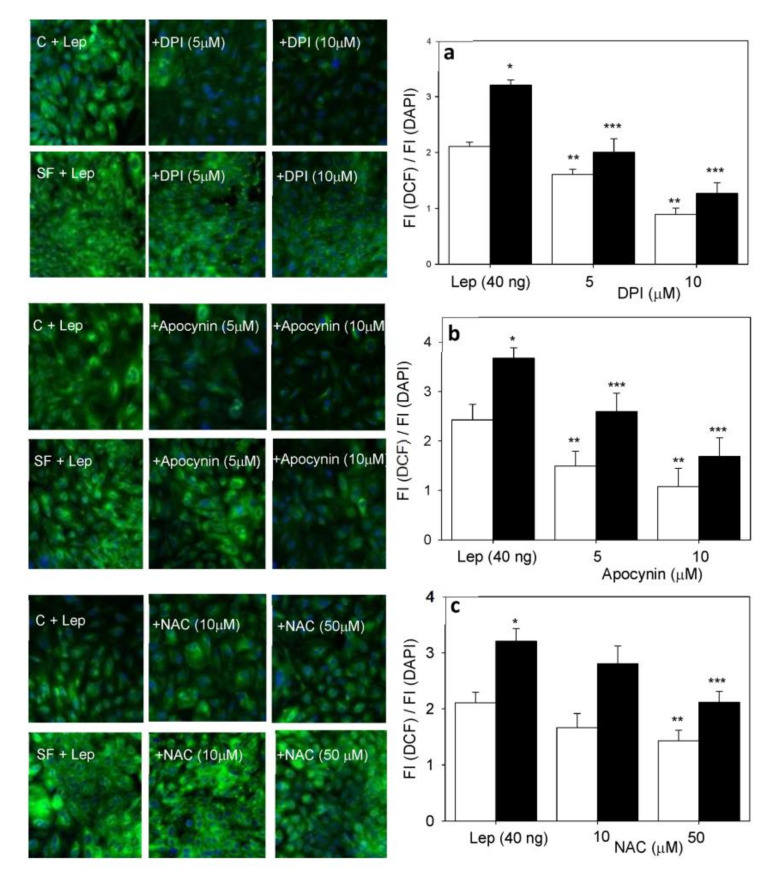
The effect of DPI, apocynin and NAC on leptin-induced ROS generation in aortic SMCs. In all panels, white bar corresponds to C cells and black bar corresponds to SF cells. the representative microscopic images of C and SF SMCs incubated with DCF (Green) and DAPI (Blue). Aortic SMCs from both C and SF rats were pre-incubated with 5 and 10 μM DPI (panel (**a**)) and apocynin (panel (**b**)) as Nox inhibitors, and with 10 and 50 μM N-acetyl-cysteine (NAC) as a reducing agent (panel (**c**)). After 30 min preincubation with inhibitors and scavenger, aortic SMCs were treated with 10 μM DCF-DA and 15 min later with leptin at 40 ng/mL culture medium for 15 more minutes. The values correspond to the ratio of the intensity of fluorescence emitted by DCF in cell (DCF FI) to the intensity of fluorescence of DAPI in cell (DAPI FI) +/− SD (n = 4 different experimental animals).* *p* < 0.05 corresponds to C + Lep vs. SF + Lep, ** *p* < 0.05 corresponds to C + Lep vs. C + Lep + DPI, *** *p* < 0.05 corresponds to SF + Lep vs. SF + Lep + DPI.

**Figure 3 antioxidants-12-00728-f003:**
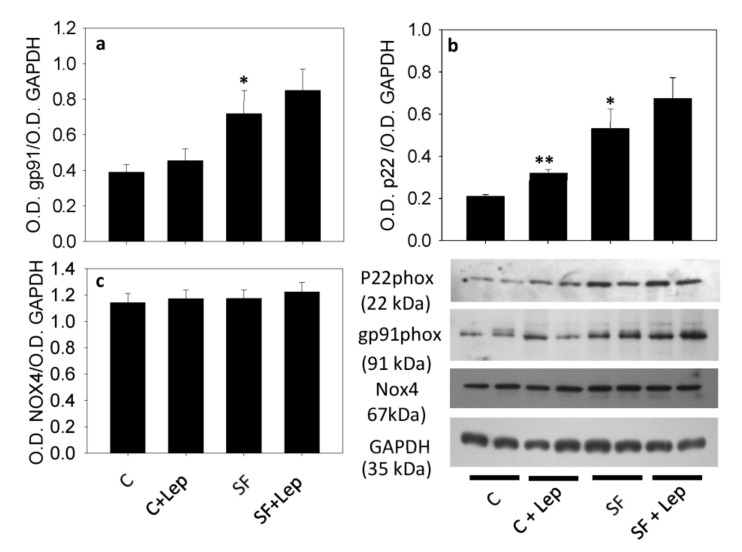
Analysis by western blot of Nox subunit proteins in aortic SMC. The subunits p22phox (22 kDa), gp91phox (91 kDa) and Nox4 (70 kDa) were detected in aortic SMCs from C and SF rats and in cells treated with leptin at 40 ng/mL culture medium for 24 h. At the end of the experiment cell were lysed and proteins were quantified according to Bradford as described in Methods section. Fifty μg protein of sample was loaded into an SDS-PAGE gel with 10% of the acrylamide for and Nox4 (70 kDa and gp91phox (91 kDa) and with 12% for p22phox (22 kDa). The graphs correspond to the ratio of optical densities (O.D.) of the above-mentioned proteins to the O.D. of GAPDH used as a control load. The panel (**a**) corresponds to the subunit gp91phox, the panel (**b**) corresponds to p22phox and the panel (**c**) corresponds to Mox4. The values correspond to the mean ratio of optical densities +/− SD (n = 4 different experimental animals).* *p* < 0.05 corresponds to C vs. SF, ** *p* < 0.005 corresponds to C vs. C + Lep.

**Figure 4 antioxidants-12-00728-f004:**
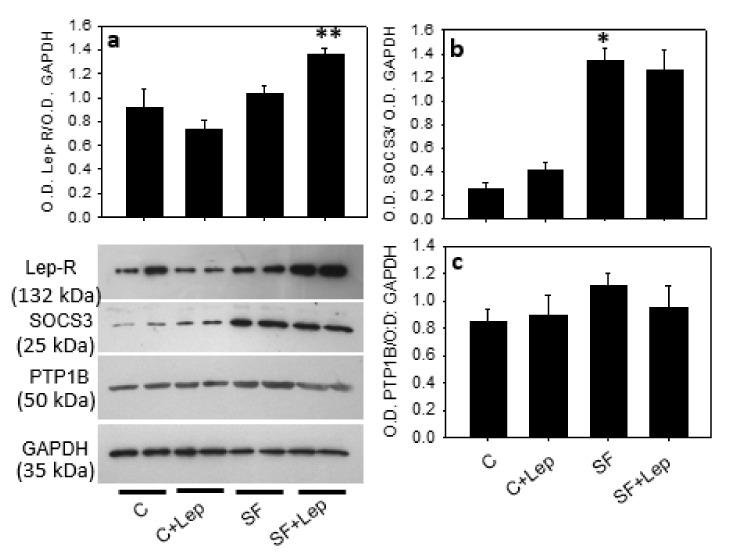
Analysis by western blot of leptin receptor (Panel (**a**)), SOCS3 (Panel (**b**)) and PTP1B (Panel (**c**)) detected in aortic SMCs from the C and SF rats and in cells treated with leptin at 40 ng/mL culture medium for 24 h. At the end of the experiment cell were lysed and proteins were quantified according to Bradford as described in Methods section. Fifty μg protein of sample was loaded into an SDS-PAGE gel with 8% of the acrylamide for leptin receptor and 10% for SOCS3 and PTP1B. The graph corresponds to the ratio of optical densities (O.D.) of proteins mentioned above and O.D. of GAPDH used as a control load. The values correspond to the mean +/− SD (n = 4 different experimental animals). * *p* < 0.05 corresponds to C vs. SF. ** *p* < 0.05 SF vs. SF + Lep.

**Figure 5 antioxidants-12-00728-f005:**
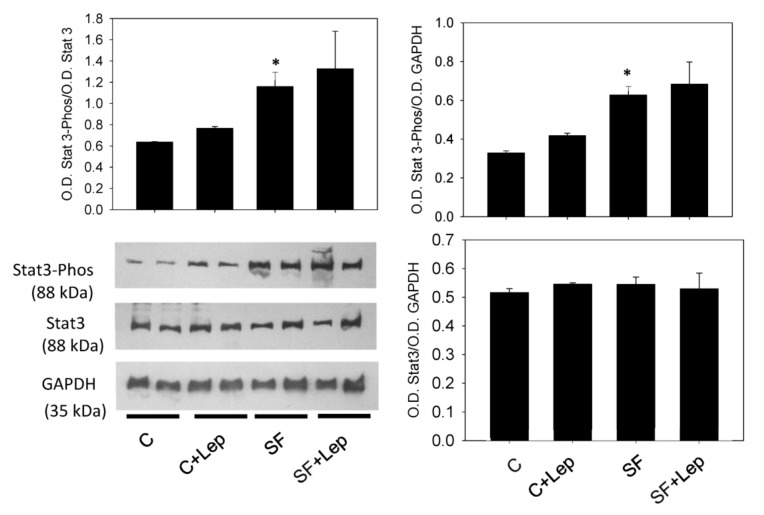
Effect of sucrose feeding and leptin treatment on STAT3 and STAT3 phosphorylation of aortic SMC. The proteins mentioned above were detected in aortic SMCs from C and SF rats and in cells treated with leptin at 40 ng/mL culture medium for 15 min stimulation. At the end of the experiment cell were lysed and proteins were quantified according to Bradford as described in Methods section. Eighty μg protein of sample was loaded into an SDS-PAGE gel with 10% of the acrylamide. The graph corresponds to the ratio of the OD of STAT3 and STAT3-phosphorylated and the OD of GAPDH used as a control load. The values correspond to the mean +/− SD (n = 4 different experimental animals). * *p* < 0.05 corresponds to C vs. SF.

**Figure 6 antioxidants-12-00728-f006:**
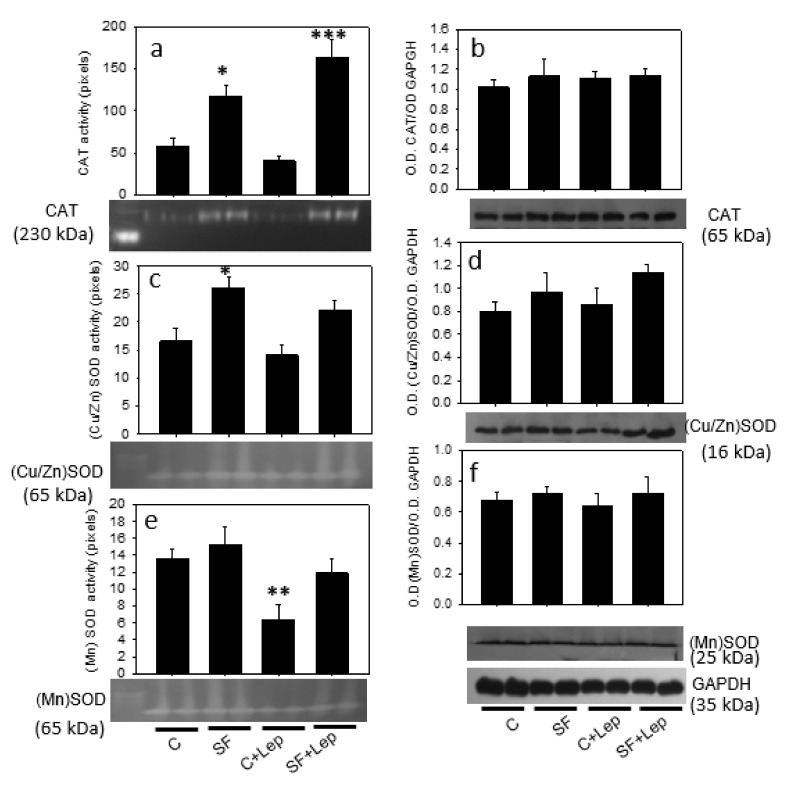
Western blot and antioxidant enzyme activities in aortic smooth muscle cells. Representative Western blot of catalase, (Cu/Zn)SOD and (Mn)SOD. Graphs (**a**,**c**,**e**) represent protein amounts of catalase (60 kDa), Cu/Zn-SOD (23 kDa) and Mn-SOD (25 kDa), respectively, with and without leptin treatment at a 40 ng/mL culture medium. Graphs (**b**,**d**,**f**) correspond to the activities of catalase, (Cu/Zn)SOD and Mn-SOD, respectively. GAPDH (37 kDa) was used as a control load. The results obtained are the mean of four independent experiments ± SD; for each protein, a representative experiment is shown (right panels). * *p* < 0.05 corresponds to C vs. SF, ** *p* < 0.005 corresponds to C vs. C + Lep, *** *p* < 0.05 corresponds to SF vs. SF + Lep.

**Table 1 antioxidants-12-00728-t001:** General characteristics of animals.

**Variables**	**C**	**SF**
Blood pressure (mm/Hg)	121.0 ± 2.2	148.2 ± 4.2 *
TGs (mM)	0.8 ± 0.1	1.8 ± 0.1 **
FFAs (mM)	0.6 ± 0.1	1.1 ± 0.1 **
Insulin (pM)	99.6 ± 5.1	167.6 ± 7.8 **
Leptin (ng/mL)	0.6 ± 0.2	2.7 ± 0.3 **
Intra-abdominal fat (g)	7.4 ± 2.2	15.3 ± 2.6 *
Body weight (g)	488.0 ± 19.1	477.0 ± 31.7
Cholesterol-HDL (mg/dL)	42.65 ± 3.5	28.1 ± 4.1 *
Glucose (mM)	5.9 ± 0.1	5.9 ± 0.2
HOMA-IR Total cholesterol (mM)	3.5 ± 0.3 1.5 ± 0.1	6.2 ± 0.8 * 1.4 ± 0.1
Plasma oxidative stress markers: TBARS (mM)	28.7 ± 6.7	43.3 ± 8.2 **
Carbonyl protein (nmol/mg protein)	3.5 ± 1.2	5.25 ± 1.7 *

Values are expressed as mean ± SD (n = 7 different animals). C, control; SF, sucrose-fed; TGs, triglycerides; FFAs, free fatty acids; TBARS, thiobarbituric acid reactive substances. The HOMA-IR was calculated from the insulin and glucose values reported in the table using the following formula: {([insulin] (in mU/L) × [glucose] (in mmol/L))/22.5}. The values of all variables were obtained at the end of the treatment period. * *p* < 0.05 SF vs. C. ** *p* < 0.01 SF vs. C.

## Data Availability

Not applicable.

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
