# Peer review of "Smooth Muscle Cells from a Rat Model of Obesity and Hyperleptinemia Are Partially Resistant to Leptin-Induced Reactive Oxygen Species Generation"

_antioxidants, 2023, doi:10.3390/antiox12030728_

Round 1

Reviewer 1 Report

There are some comments shows follows for readers.

1. Why did body weight and blood glucose levels not change between control and sucrose-fed group? It is difficult to say the animal model of sucrose feeding useful.

2. Is there any histological changes in aorta after feeding sucrose?

3. How leptin control catalase, Cu/Zn SOD, and Mn SOD activity? What mechanisms may involve in this?

Author Response

Reviewer I

Comment 1. Why did body weight and blood glucose levels not change between control and sucrose-fed group? It is difficult to say the animal model of sucrose feeding useful.

Answer: The lack of differences in weight gain between the two groups may be due to the lack of differences in energy intake. In a paper published by our group (El Hafidi et al. 2004), we monitored food, drinking water and energy intake, and observed that SF rats ingested less solid food than control rats, which contributed to a lower energy intake from solid food. These results allowed us to propose that decreased solid food intake by SF rats influences weight gain although exposure to liquid sucrose contributed to increased and accumulated body fat in the intra-abdominal cavity. Thus, we think that the fact that sucrose intake leads to an increase in body fat may provide useful information on the effects of diet on the metabolic homeostasis of the vascular smooth muscle cell.

A comment about this finding was added in the discusuion section  (see page: Line

El Hafidi, M., Pérez, I., Zamora, J., Soto, V., Carvajal-Sandoval, G., & Baños, G. (2004). Glycine intake decreases plasma free fatty acids, adipose cell size, and blood pressure in sucrose-fed rats. American journal of physiology. Regulatory, integrative and comparative physiology287(6), R1387–R1393. https://doi.org/10.1152/ajpregu.00159.2004

Comment 2. Is there any histological changes in aorta after feeding sucrose?

Answer: At our National Institute of Cardiology, a group of researchers using the same sucrose rat model for other purposes has recently demonstrated a significant structural change of the media layer of the thoracic aorta of sucrose-fed animals compared to control animals (Castrejón-Téllez et al 2022). The group proposes that sucrose diet induces a change in vascular tissue thickness that may be associated with a change in the contractile to synthetic (proliferative) smooth muscle cell phenotype that is associated with alterations in aortic ring vascular reactivity and hypertension.

Castrejón-Téllez V, Rubio-Ruiz ME, Cano-Martínez A, Pérez-Torres I, Del Valle-Mondragón L, Carreón-Torres E, Guarner-Lans V. High Sucrose Ingestion during a Critical Period of Vessel Development Promotes the Synthetic Phenotype of Vascular Smooth Muscle Cells and Modifies Vascular Contractility Leading to Hypertension in Adult Rats. Int J Hypertens. 2022 Jun 21;2022:2298329. doi: 10.1155/2022/2298329. PMID: 35774422; PMCID: PMC9239805

Comment 3: How leptin control catalase, Cu/Zn SOD, and Mn SOD activity? What mechanisms may involve in this?

Answer: To our current knowledge, there is no data about the direct effect of leptin on the activity of antioxidant enzymes (catalase, Cu/Zn SOD and Mn SOD). In smooth muscle cells, leptin may regulate antioxidant defenses in response to increased reactive oxygen species. Under conditions of oxidative stress induced by high ROS generation, the increased expression of antioxidant enzymes has been described in skeletal muscle (Franco, A. A., Odom, R. S., & Rando, T. A. (1999). Regulation of antioxidant enzyme gene expression in response to oxidative stress and during differentiation of mouse skeletal muscle. Free radical biology & medicine27(9-10), 1122–1132. https://doi.org/10.1016/s0891-5849(99)00166-5.). However, the endogenous oxidant such as H2O2 system is considered of central importance in redox signaling as an efficient oxidant agent to modulate cellular redox status including enzyme activities. ROS at elevated concentrations can react with the catalytic site and other amino acid residues to induce conformational change of the proteins and reduce or inhibit the activity of antioxidant enzymes. In control aortic SMCs, leptin-reduced effect of Mn-SOD activity can be indirectly related through its ROS-inducing effect via activation of NADPH oxidase and mitochondria  [Yamagishi , S. I., Edelstein, D., Du, X. L., Kaneda, Y., Guzmán, M., & Brownlee, M. (2001). Leptin induces mitochondrial superoxide production and monocyte chemoattractant protein-1 expression in aortic endothelial cells by increasing fatty acid oxidation via protein kinase A. The Journal of biological chemistry276(27), 25096–25100. https://doi.org/10.1074/jbc.M007383200,  López-Acosta, O., Ruiz-Ramírez, A., Barrios-Maya, M. Á., Alarcon-Aguilar, J., Alarcon-Enos, J., Céspedes Acuña, C. L., & El-Hafidi, M. (2023). Lipotoxicity, glucotoxicity and some strategies to protect vascular smooth muscle cell against proliferative phenotype in metabolic syndrome. Food and chemical toxicology : an international journal published for the British Industrial Biological Research Association172, 113546. https://doi.org/10.1016/j.fct.2022.113546)]Therefore, further studies are needed to elucidate the mechanism by which leptin reduce the Mn-SOD activity in SMC from control animals.

Reviewer 2 Report

The study by Lopez-Acosta et al. entitled, “Smooth muscle cells from rat model of obesity and hyperleptinemia are partially resistance to leptin-induced reactive oxygen species generation” sought to evaluate the effect of leptin on reactive oxygen species in smooth muscle cells from a rat model of obesity and hyperleptinemia. Although this study has potential biomedical relevance, and investigators used primary vascular smooth muscle cells from control and sucrose-supplemented animals (to be commended). there are issues that limit the manuscript in its current form. The comments provided below are intended to improve the overall scientific merit of the manuscript.

-       English language editing required

-       The title states “rat model of obesity” but there were no differences in body weight between groups.  Did authors collect data on lean mass? If fat mass was ~2-fold higher but there wasn’t a difference in bodyweight, what is accounting for this? Intra-abdominal fat is reported, but what about subcutaneous fat?

-       Concentration of leptin in cell culture experiments should be included in the methods. How do these concentrations compare to circulating hyperleptinemia values?

-       How was blood pressure assessed? Not included in the methods

-       Sucrose-supplemented mice likely have insulin resistance given the fasting glucose and insulin values. It would be helpful from a clinical translational perspective if authors would calculate HOMA-IR

-       Do leptin receptor deficient mice have elevated cardiovascular/vascular dysfunction? These results should be considered when interpreting the results of the present study

-       All figure legends should include the tissue/cell type that was analyzed

Author Response

Reviewer II

Comment:  English language editing required

Answer: the English language was edited see manuscrit in red version . 

Comment: The title states “rat model of obesity” but there were no differences in body weight between groups.  Did authors collect data on lean mass? If fat mass was ~2-fold higher but there wasn’t a difference in bodyweight, what is accounting for this? Intra-abdominal fat is reported, but what about subcutaneous fat?

Answer: The title stated that sucrose-fed rats were a model of obesity because obesity has also been defined as the presence of excess adipose tissue and not just weight gain [1].  In a previous work by our group (El Hafidi et al. 2004), we described that the lack of weight difference between the control and sucrose fed groups can be attributed to the lack of difference in their energy intake. Because the sucrose-fed rats ingested less solid food than the control animals.

Body composition was not examined in detail in this study, and we do not know whether sucrose administration caused an increase in adipose tissue content at the expense of other tissues. The SF rats consumed approximately half the amount of food consumed by animals that were not given sucrose; consequently, the availability of nutrients from the solid food was lower as described previously (el Hafidi et al 2004). Thus, the lower energy intake of the SF rats was compensated by the additional calories from the sucrose solution.

El Hafidi M, Pérez I, Zamora J, Soto V, Carvajal-Sandoval G, Baños G. Glycine intake decreases plasma free fatty acids, adipose cell size, and blood pressure in sucrose-fed rats. Am J Physiol Regul Integr Comp Physiol. 2004 Dec;287(6):R1387-93. doi: 10.1152/ajpregu.00159.2004. Epub 2004 Aug 26. PMID: 15331379.

Comment:  Concentration of leptin in cell culture experiments should be included in the methods. How do these concentrations compare to circulating hyperleptinemia values?

Answer: The concentrations of leptin used in all assays were included in all figure legends.

As for the cell culture experiments, the concentrations of leptins used in the experiments far exceed the concentrations found for plasma leptin from sucrose animals. Prior to testing the 20, 40 and 80 ng concentrations, concentrations below 20 ng or similar to physiological concentrations were investigated but no changes in the fluorescence index of ROS release were observed in cells extracted from control animals.

Comment:  How was blood pressure assessed? Not included in the methods

Answer: The brief description of the method of blood pressure assessment in rats cited below, is added to the Method section.

 At the end of the treatment period, measurements of the rat’s blood pressure were taken by the tail-cuff method: the cuff was connected to a pneumatic pulse transducer (Narco Bio-Systems from Healthdyne Co) and a Programmed Electro-Sphygmomanometer from the same company. The recordings were obtained in duplicate by means of a Narco Bio-Systems polygraph. As described by El Hafidi et al. 1997)

Comment:  Sucrose-supplemented mice likely have insulin resistance given the fasting glucose and insulin values. It would be helpful from a clinical translational perspective if authors would calculate HOMA-IR

Answer: The HOMA-IR was calculated using the insulin and glucose values reported in the manuscript (Table 1) by applying the following formula: {([insulin] (in mU/l) x [glucose] (in mmol/l))/22.5}. The results were added to the general characteristics of the animals (Table 1). The results show a significant difference in HOMA-IR between the control and sucrose model indicated insulin resistance in the model as demonstrated by the hyperinsulinemia/euglycemia clamp experiment reported previously published paper (El Hafidi et al. 2018) where sucrose rats were shown to develop insulin resistance. This comment is added to Result section of the the manuscript

Comment:  Do leptin receptor deficient mice have elevated cardiovascular/vascular dysfunction? These results should be considered when interpreting the results of the present study

Answer: A flowing comment about this observation was added in the Discussion section.

There are several articles describing that leptin receptor-deficient ob/ob and db/db mice develop cardiovascular and vascular dysfunction (Koh, K. K., Park, S. M., & Quon, M. J. (2008). Leptin and cardiovascular disease: response to therapeutic interventions. Circulation117(25), 3238–3249. https://doi.org/10.1161/CIRCULATIONAHA.107.741645). These leptin and leptin receptor-deficient rodent models have provided many useful insights into the underlying molecular and pathophysiological mechanisms of metabolic and cardiovascular diseases associated with obesity and type 2 diabetes. For example, it has been described that animals deficient in leptin receptors show elevated triglyceride levels and lipid accumulation in the myocardium, which may promote lipotoxicity and directly impacts cardiac contractility. (Poetsch, M. S., Strano, A., & Guan, K. (2020). Role of Leptin in Cardiovascular Diseases. Frontiers in endocrinology11, 354. https://doi.org/10.3389/fendo.2020.00354).   

Comment  All figure legends should include tissues and cell that was analyzed

Answer :The aortic smooth muscle cell was included in all figure legend.

Round 2

Reviewer 2 Report

Thank you for your attention to my previous comments. I have no further comments.